# Morphological and ultrastructural investigation of the posterior atlanto-occipital membrane: Comparing children with Chiari malformation type I and controls

Vijay M. Ravindra[1,2,3,4]ʘ*, Lorraina Robinson[5]ʘ, Hailey Jensen[6], Elena Kurudza[1], Evan Joyce[1], Allison Ludwick[2], Russell Telford[6], Osama Youssef[7], Justin Ryan[3,4], Robert J. Bollo[1,2], Rajiv R. Iyer[1,2], John R. W. Kestle[1,2], Samuel H. Cheshier[1,2,7], Daniel S. Ikeda[8], Qinwen Mao[5], Douglas L. Brockmeyer[1,2]

1 Department of Neurosurgery, Clinical Neurosciences Center, University of Utah, Salt Lake City, Utah, United States of America, 2 Division of Pediatric Neurosurgery, Primary Children's Hospital, Salt Lake City, Utah, United States of America, 3 Department of Neurosurgery, University of California San Diego, San Diego, California, United States of America, 4 Division of Pediatric Neurosurgery, Rady Children's Hospital, San Diego, California, United States of America, 5 Department of Pathology, University of Utah, Salt Lake City, Utah, United States of America, 6 Department of Pediatrics, University of Utah, Data Coordinating Center, Salt Lake City, Utah, United States of America, 7 Huntsman Cancer Institute, Salt Lake City, Utah, United States of America, 8 Walter Reed National Military Medical Center, Bethesda, Maryland, United States of America

ʘ These authors contributed equally to this work.
* vijay.ravindra@hsc.utah.edu

**Data Availability Statement:** All relevant data are within the paper and its Supporting information files.

## Abstract

### Introduction

The fibrous posterior atlanto-occipital membrane (PAOM) at the craniocervical junction is typically removed during decompression surgery for Chiari malformation type I (CM-I); however, its importance and ultrastructural architecture have not been investigated in children. We hypothesized that there are structural differences in the PAOM of patients with CM-I and those without.

### Methods

In this prospective study, blinded pathological analysis was performed on PAOM specimens from children who had surgery for CM-I and children who had surgery for posterior fossa tumors (controls). Clinical and radiographic data were collected. Statistical analysis included comparisons between the CM-I and control cohorts and correlations with imaging measures.

### Results

A total of 35 children (mean age at surgery 10.7 years; 94.3% white) with viable specimens for evaluation were enrolled: 24 with CM-I and 11 controls. There were no statistical demographic differences between the two cohorts. Four children had a family history of CM-I and five had a syndromic condition. The cohorts had similar measurements of tonsillar descent,

**Funding:** The American Syringomyelia & Chiari Alliance Project Inc. (ASAP) funded the project for a sum of $19,982.00 as part of the Timothy M. George Fellowship Award to Vijay Ravindra.

**Competing interests:** The authors have declared that no competing interests exist.

**Abbreviations:** C-C2SVA, condylar-C2 sagittal vertical alignment; CCJ, craniocervical junction; CM-I, Chiari I malformation; CSF, cerebrospinal fluid; CXA, clival-axial angle; MRI, magnetic resonance imaging; PAAM, posterior atlantoaxial membrane; PAOM, posterior atlanto-occipital membrane; pBC2, maximum perpendicular distance to the basion-inferior aspect of the C2 body.

syringomyelia, basion to C2, and condylar-to-C2 vertical axis (all p>0.05). The clival-axial angle was lower in patients with CM-I (138.1 vs. 149.3 degrees, p = 0.016). Morphologically, the PAOM demonstrated statistically higher proportions of disorganized architecture in patients with CM-I (75.0% vs. 36.4%, p = 0.012). There were no differences in PAOM fat, elastin, or collagen percentages overall and no differences in imaging or ultrastructural findings between male and female patients. Posterior fossa volume was lower in children with CM-I (163,234 mm$^3$ vs. 218,305 mm$^3$, p<0.001), a difference that persisted after normalizing for patient height (129.9 vs. 160.9, p = 0.028).

## Conclusions

In patients with CM-I, the PAOM demonstrates disorganized architecture compared with that of control patients. This likely represents an anatomic adaptation in the presence of CM-I rather than a pathologic contribution.

## Introduction

Chiari malformation type I (CM-I) is defined as the displacement of the cerebellar tonsils $\geq 5$ mm below the foramen magnum [1]. Children with CM-I can be asymptomatic or can have severe symptoms that profoundly impact quality of life. Among these, CM-I can be associated with the development of a cavity within the spinal column (syringomyelia) [2, 3], which can lead to severe neurological impairment. Surgical treatment for patients with CM-I with or without syringomyelia is undertaken to expand the posterior fossa to relieve cerebrospinal fluid (CSF) obstruction, equalize the craniospinal pressure gradient, and allow for normal physiologic CSF flow at the foramen magnum.

The posterior atlanto-occipital membrane (PAOM), also referred to as the dural band or thickened dura mater, is a fibrous structure located dorsal to the dura at the posterior cranio-cervical junction (CCJ) that stretches between the bilateral occipital condyles and upper borders of the C1 lamina (Fig 1). It is encountered and removed along with the occipital bone during first-time surgical approaches to the CCJ and posterior cranial fossa for decompression of a Chiari malformation or removal of a posterior fossa tumor.

Alabaster et al. [4] studied the histological characteristics of this membrane in adults using 10 cadaveric specimens and 39 patients—31 with CM-I and 8 with other posterior fossa pathologies. They found that the ligament in patients with CM-I was disorganized, with poorly arranged collagen bands, interspersed adipose tissue, hyaline nodules present, and altered fiber orientation, indicating pathological differences in the PAOM of adults with CM-I from those with other pathologies. A similar histological analysis has not been performed in children. The use of magnetic resonance imaging (MRI), of either the brain or the cervical spine, is common when evaluating children with CM-I but cannot be used to evaluate specific characteristics of the PAOM. Further evaluation and understanding of this structure and its potential contribution to the pathophysiology of CM-I is necessary to assess the success of bone-only decompression.

In this study, we aimed to perform a detailed histological/morphological analysis of the PAOM in children with CM-I and control patients. We also aimed to determine whether there are differences in commonly used imaging parameters for the CCJ between patients with CM-I and those without the condition. We hypothesized that there are structural differences in the PAOM and the imaging measurements of these two pediatric cohorts.

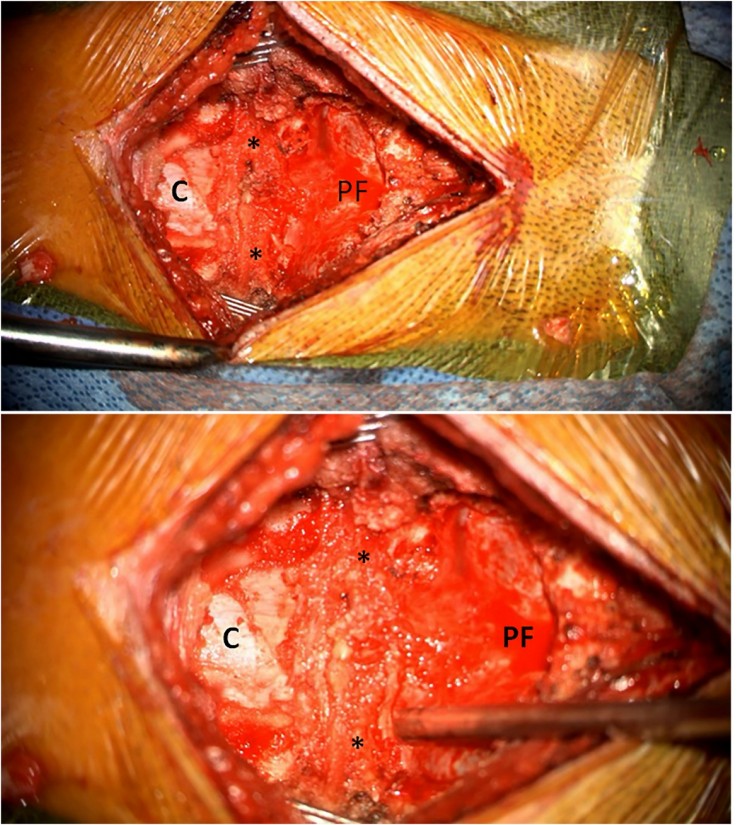

**Fig 1. Intraoperative surgical view of low (top) and high (bottom) magnification following C1 laminectomy, suboccipital craniectomy in preparation for Chiari decompression surgery.** C denotes cervical dura, PF indicates posterior fossa dura, * indicates the lateral aspects of the PAOM.

## Materials and methods

### Study population

This was a prospective pathological and clinical study conducted at Primary Children's Hospital under the joint University of Utah/Primary Children's Hospital institutional review board approval (protocol #00144341) with written informed consent from the parents or guardians of the participants. Enrollment began on September 1, 2021, and ended June 30, 2023. Included children were all <18 years of age. The experimental group comprised children undergoing surgical treatment of CM-I and the control group comprised children requiring surgical excision of a posterior fossa tumor. Exclusion criteria included revision or redo surgery or lack of consent for tissue analysis/collection. All patients were operated on by fellowship-trained pediatric neurosurgeons. For this study, the PAOM of each participating child was evaluated pathologically and radiologically.

### Surgical technique

The PAOM is removed during the routine surgical treatment of CM-I or posterior fossa brain tumor. Surgical excision is undertaken using a standard midline skin incision that extends from the external occipital protuberance to the C2 spinous process. The incision is carried through the nuchal ligament using Bovie electrocautery. The suboccipital muscles are then

dissected and retracted with self-retaining retractors. Standard suboccipital craniectomy and C1 laminectomy are performed using a high-speed drill. The PAOM is identified and resected sharply in one piece several millimeters medial to the ligament's origin at the occipital condyle and is detached from the underlying dura. The goal is to remove the specimen with 0.5–1 cm on both sides of the midline.

After harvesting, each specimen was placed in 10% formalin and transported to the pathology lab for cutting and staining of the specimen.

## Pathological analysis

Standardized pathological evaluation for each specimen included cutting, sectioning longitudinally, and cross sectioning at the edges. Specimens were processed and placed into paraffin blocks. Tissue sections (4 μm thick) were cut from the paraffin blocks and applied to glass slides. Hematoxylin and eosin and Masson's trichrome staining were performed. Each of the sections was evaluated for maximum thickness (mm), hyaline nodules, calcifications, ossification, adipose tissue content, and fibrous splitting of the collagen fibers.

Blinded histological examination was performed using routine microscopy by a fellowship-trained, attending neuropathologist. Additional staining included immunohistochemistry studies: smooth muscle antigen staining for myofibroblasts, epithelial membrane antigen staining for meningothelial cells, and type IV collagen staining to determine overall connective tissue morphology.

The resultant pathological characteristics used for comparison included percentage of collagen, percentage of elastin, and fat content. A disorganization scoring system was used to describe the organization of the membrane using four categories, 0 to 3, representing increasing levels of disorientation of collagen bands and more interspersed adipose tissue (Fig 2).

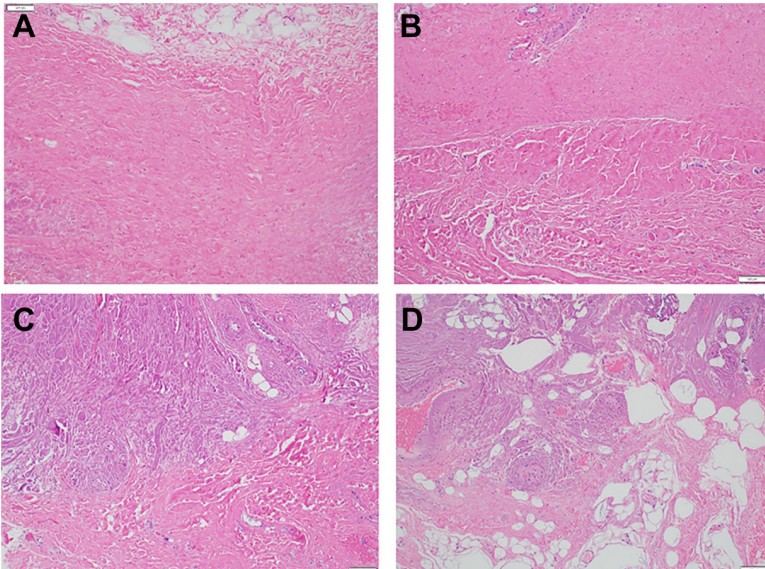

**Fig 2. Hematoxylin and eosin stains demonstrating the disorganization score used for analysis of the PAOM.** (A) Disorganization score of zero, with parallel collagen bands with no interspersed adipose tissue. (B) Disorganization score of one, demonstrating collagen bands with altered orientation focally with no interspersed adipose tissue. (C) Disorganization score of two, with collagen bands with altered orientation diffusely with small amount of interspersed adipose tissue. (D) Disorganization score of three, with poorly arranged collagen bands with significant amount of interspersed adipose tissue.

Each specimen was independently reviewed by two fellowship-trained neuropathologists who tested the method and consistently arrived at consensus. Microscopic thickness of the membrane was also recorded (μm).

## Clinical information

Baseline demographic and clinical information collected for each patient enrolled included sex, age at surgery, race, height, weight, and body mass index (weight (kg)/height (m)$^2$). Family history of CM-I and presence of syndromic conditions were also recorded. Presenting signs and symptoms were recorded, including posterior neck pain, sensory symptoms, ataxia, motor weakness, dizziness/lightheadedness, visual changes, cognitive deficits, and swallowing difficulty.

Surgery-related complications, including CSF leak, surgical site infection, pseudomeningocele, and meningitis, as well as reoperation and readmission were also recorded.

## Radiographic measurements

Radiographic variables were recorded from MRI scans obtained before treatment. Imaging characteristics included the presence of syrinx, tonsillar descent (mm), condylar-to-C2 sagittal vertical axis (C-C2SVA) [5] measured in mm (Fig 3A), maximum perpendicular distance to the basion-inferior aspect of the C2 body (pBC2) measured in mm (Fig 3B), and the clival-axial angle (CXA) measured in degrees (Fig 3C). Additional measurements included radiographic thickness of the PAOM and the posterior atlantoaxial membrane (PAAM) measured in mm (Fig 3C).

## Posterior fossa volume

Three-dimensional reconstructions of the posterior fossa were performed in Mimics Innovation Suite (version 25, Leuven, Belgium) to calculate volume. T1-weighted MRI series were imported into the software where the brainstem, cerebellum, and surrounding fluid were reconstructed. The brainstem was truncated on the superior aspect by creating a line between the superior aspect of the tentorium and the superior aspect of the pons; segmentation elements above this line were removed from the 3D reconstruction. The region below the foramen magnum was truncated regardless of cerebellar tissue extension inferior to the foramen magnum to most accurately measure the compartment available for CSF, brain, and vascular

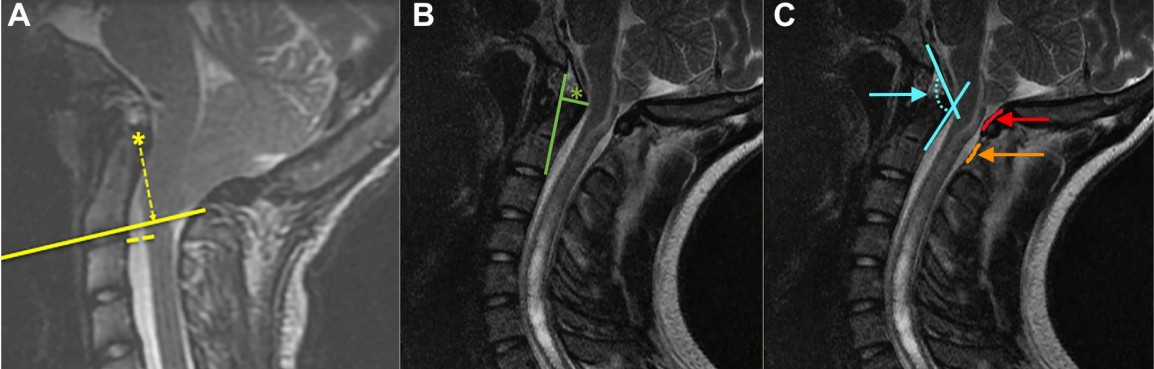

**Fig 3. Steps demonstrating the measurement of the C-C2SVA (A), pBC2 (B), and CXA (in light blue), PAOM (in red), and PAAM (in orange) (C).** (Panel A reproduced with permission from Ravindra et al. [5]).

contents. Fluid and other soft tissue described by the same boundaries within the cranium were included. The resulting segmentation defined the extent of the posterior fossa for this study. The medical image reconstruction software derived a volume from this segmented region. An example of the technique is illustrated in Fig 4.

The posterior fossa volume calculations obtained were treated independently and indexed to patient height ((posterior fossa volume in mm$^3$)/height in mm).

## Statistical analysis

Descriptive statistics of PAOM specimens were summarized as means and standard deviations for continuous variables, and counts and percentages for categorical variables. Demographic, clinical, and radiographic comparisons between PAOM specimens from CM-I patients and controls were made using Fisher's exact tests and Wilcoxon rank-sum tests. Associations between imaging and pathology measurements in the PAOM specimens were also assessed. Male and female specimens were compared as well. The Pearson correlation coefficient was used to analyze the relationship between imaging and pathology measurements in PAOM specimens. All analyses were conducted using SAS 9.4 (SAS Institute, Cary, North Carolina, USA).

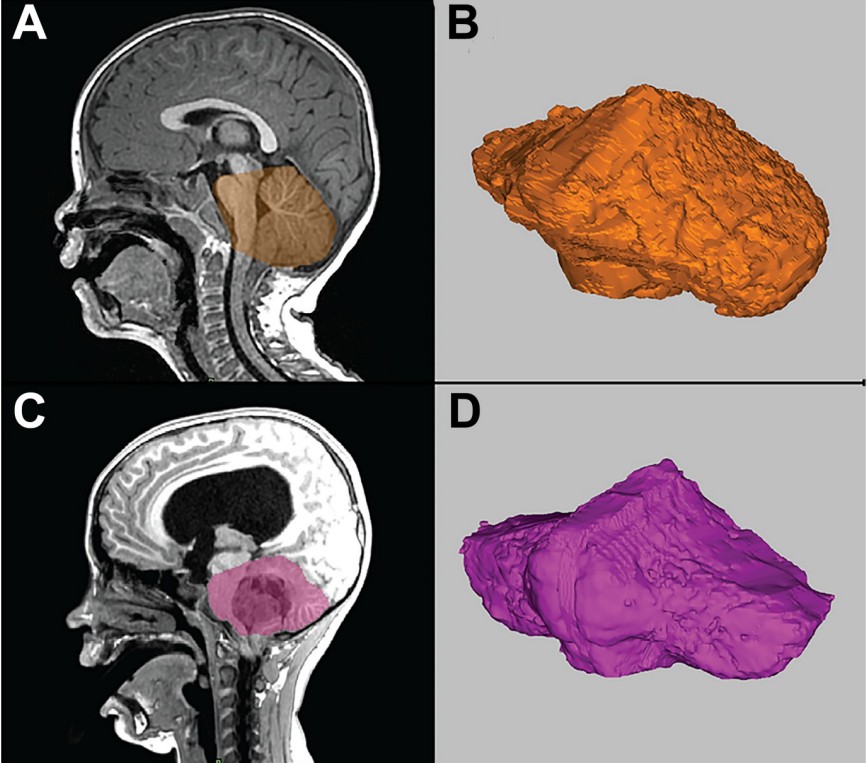

**Fig 4. Demonstration of the calculation of the three-dimensional volume of the posterior fossa using sagittal high-resolution T1-weighted MRI (A, C) with three-dimensional volumetric reconstruction (B, D).** Panels A and B show a patient with Chiari I malformation. Panels C and D show a patient with posterior fossa brain tumor.

## Results

### Demographics

A total of 35 children were enrolled into the study (Fig 5). Specimens were obtained from 24 patients with CM-I and 11 control patients (S1 Data). The demographic characteristics of the two groups were similar (Table 1). Most of the children were white (94.3%), and 15 participants were female (42.9%). The mean age at surgery was 128.3 months (10.7 years) and did not differ between patients with CM-I and controls (136.8 vs. 110.0 months, p = 0.195). No differences were seen in height, weight, or body mass index. Four children with CM-I had a confirmed family history of the condition. Overall, five children had concomitant syndromic conditions (14.3%).

### Clinical presentation

Overall, most children presented with headaches (82.9%) (Table 2). Four children with CM-I presented with posterior neck pain. Children with posterior fossa brain tumors more commonly presented with ataxia (45.5% vs. 8.3%, p = 0.011). Similar rates of motor weakness, dizziness/lightheadedness, visual changes, and cognitive deficits were seen among patients with CM-I and controls (Table 2). Children with CM-I had a nonsignificantly higher rate of swallowing difficulty (25.0% vs. 9.1%, p = 0.282).

### Surgical complications

No children experienced CSF leak after surgery. Two children in the CM-I cohort experienced surgical site infection and were treated with oral antibiotics after clinical evaluation. Two children in the CM-I cohort required readmission and reoperation for persistent symptoms/syringomyelia (Table 3).

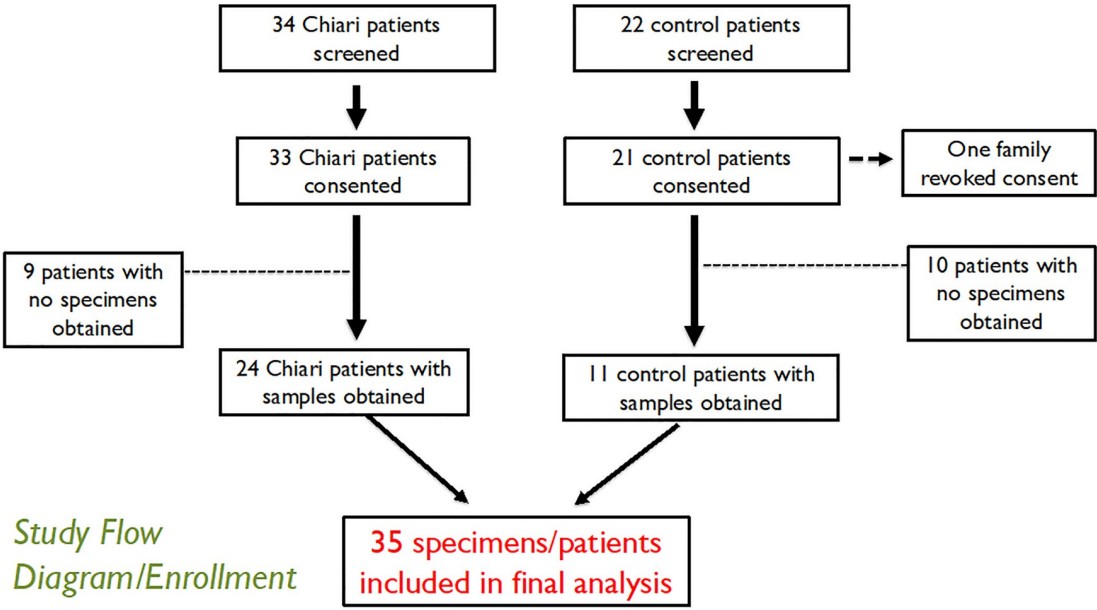

**Fig 5. CONSORT diagram describing screening and enrollment in the study.**

**Table 1. Demographic information for Chiari I malformation and control cohorts.**

| Variable | Chiari (n = 24) | Control (n = 11) | Overall (n = 35) | p-value |
|---|---|---|---|---|
| Race | | | | 0.092[a] |
| White | 24 (100.0%) | 9 (81.8%) | 33 (94.3%) | |
| Other | 0 (0.0%) | 2 (18.2%) | 2 (5.7%) | |
| Female sex | 12 (50%) | 3 (27.3%) | 15 (42.9%) | 0.281[a] |
| Age at surgery (months) | 136.8 (64.80) | 110.0 (52.92) | 128.3 (61.83) | 0.195[b] |
| | 169.5 [73.5, 190.0] | 84.0 [59.0, 171.0] | 131.0 [71.0, 186.0] | |
| Height (m) | 1.4 (0.28) | 1.4 (0.30) | 1.4 (0.28) | 0.749[b] |
| | 1.5 [1.2, 1.6] | 1.3 [1.1, 1.7] | 1.5 [1.1, 1.6] | |
| Weight (kg) | 48.2 (30.46) | 35.7 (19.17) | 44.2 (27.75) | 0.424[b] |
| | 46.2 [22.6, 60.8] | 24.9 [18.3, 59.0] | 43.7 [19.2, 59.9] | |
| BMI | 22.0 (8.36) | 17.7 (2.43) | 20.6 (7.29) | 0.207[b] |
| | 18.7 [16.1, 24.5] | 17.0 [15.2, 20.3] | 17.8 [16.1, 21.8] | |
| Family history of CM-I | | | | |
| Yes | 4 (16.7%) | 0 (0.0%) | 4 (11.4%) | |
| Unknown | 9 (37.5%) | 10 (90.9%) | 19 (54.3%) | |
| Syndromic conditions | 4 (16.7%) | 1 (9.1%) | 5 (14.3%) | 1.000[a] |

BMI, body mass index; CM-I, Chiari malformation I

[a] Fisher's exact test.

[b] Wilcoxon rank-sum test.

## Imaging characteristics

Similar rates of syringomyelia were seen (20.8% CM-I vs. 18.2% controls, p = 0.828) (Table 4). The mean degree of tonsillar descent was not significantly different in the CM-I cohort (12.5 mm vs. 7.8 mm, p = 0.130). Although the mean C-C2SVA was 1.5 mm greater in the CM-I cohort (4.8 mm vs. 3.3 mm, p = 0.291) and the proportion of children with CM-I with a C-C2SVA ≥5 mm was higher (45.8% vs. 36.4%, p = 0.594), the differences were not statistically significant. The pBC2 was similar in the two groups (7.4 mm vs. 7.1 mm, p = 0.418), and a nonsignificantly higher proportion of children with CM-I had a pBC2 ≥9 mm (37.5% vs. 18.2%, p = 0.349). The mean CXA was significantly lower in patients with CM-I (138.1 vs.

**Table 2. Clinical presentation of Chiari I malformation and control cohorts.**

| Variable | Chiari (n = 24) | Control (n = 11) | Overall (n = 35) | p-value[a] |
|---|---|---|---|---|
| Headache | 21 (87.5%) | 8 (72.7%) | 29 (82.9%) | 0.249 |
| Posterior neck pain | 4 (16.7%) | 0 (0.0%) | 4 (11.4%) | 0.183 |
| Sensory symptoms | 8 (33.3%) | 2 (18.2%) | 10 (28.6%) | 0.337 |
| Ataxia | 2 (8.3%) | 5 (45.5%) | 7 (20.0%) | 0.011 |
| Motor weakness | 1 (4.2%) | 1 (9.1%) | 2 (5.7%) | 0.314 |
| Dizziness/lightheaded | 8 (33.3%) | 3 (27.3%) | 11 (31.4%) | 0.855 |
| Visual changes | 2 (8.3%) | 0 (0.0%) | 2 (5.7%) | 0.768 |
| Cognitive deficit | 3 (12.5%) | 0 (0.0%) | 3 (8.6%) | 0.382 |
| Swallowing difficulty | 6 (25.0%) | 1 (9.1%) | 7 (20.0%) | 0.282 |

[a]Fisher's exact test with Lancaster's mid-p correction

Table 3. Surgical treatment complications in the Chiari I malformation and control cohorts.

| Variable | Chiari (n = 24) | Control (n = 11) | p-value[a] |
|---|---|---|---|
| CSF leak | 0 (0%) | 0 (0%) | N/A |
| Surgical site infection | 2 (8.3%) | 0 (0.0%) | 0.768 |
| Pseudomeningocele | 0 (0%) | 0 (0%) | N/A |
| Meningitis | 0 (0%) | 0 (0%) | N/A |
| Reoperation | 2 (8.3%) | 0 (0.0%) | 0.768 |
| Readmission | 2 (8.3%) | 0 (0.0%) | 0.768 |

CSF, cerebrospinal fluid; N/A, not applicable

[a]Fisher's exact test with Lancaster's mid-p correction

149.3 degrees, p = 0.016); three children with CM-I had a CXA <125 degrees. On preoperative MRI, the PAOM (2.7 mm vs. 2.8 mm, p = 0.740) and PAAM (2.6 mm vs. 3.4 mm, p = 0.376) had similar thicknesses in the CM-I and control cohorts (Table 4).

## Pathological characteristics

The overall collagen percentage was not significantly lower in the CM-I cohort (77.1% vs. 85.5%, p = 0.251) (Table 4). There was limited presence of elastin in the samples from both cohorts, and the amount did not differ between them. There was a similar fat content in the PAOM samples (22.9% vs. 14.5%, p = 0.251). With respect to disorganization scoring, 75.0% of CM-I specimens demonstrated moderate/severe disorganization versus 36.4% of control patients (p = 0.012). Microscopically, the PAOM was thinner in the CM-I cohort, but this difference was not statistically significant (2031.3 μm vs. 2363.6 μm, p = 0.142). No fiber splitting or hyaline nodules were seen in any of the specimens.

To understand any difference in tissue characteristics by sex, we compared male and female patients with CM-I (Table 5). No differences in the imaging or histopathological characteristics were discovered, although the PAOM was nonsignificantly thinner in males than in females (1895.8 μm vs. 2166.7 μm, p = 0.214).

## Imaging and pathological correlations

No significant correlations were found between C-C2SVA and the pathological results of disorganization score (r = 0.142), membrane thickness (r = 0.147), or imaging measures of PAOM thickness (r = 0.097) (Fig 6). Additionally, no significant correlations were found between CXA and the pathological results of disorganization score (r = 0.350), membrane thickness (r = -0.492), or imaging measures of PAOM thickness (r = 0.006) (Fig 7). Finally, no significant correlations were found between pBC2 and the pathological results of disorganization score (r = -0.355), membrane thickness (r = 0.285), or imaging measures of PAOM thickness (r = 0.287) (Fig 8).

## Posterior fossa volume measurements

Posterior fossa volume calculations using a 3D segmentation technique were obtained in 24 children; 11 did not have volumetric imaging that could be analyzed. Values can be seen in Table 4. Posterior fossa volume was lower in children with CM-I (163,234 mm³ vs. 218,305 mm³, p<0.001), a difference that persisted after normalizing for patient height (129.9 vs. 160.9, p = 0.028).

**Table 4. Imaging and pathological/morphological characteristics in Chiari I malformation and control cohorts.**

| Variable | Chiari (n = 24) | Control (n = 11) | p-value |
|---|---|---|---|
| **Imaging Characteristics** | | | |
| **Presence of syrinx** | 5 (20.8%) | 2 (18.2%) | 0.828[a] |
| **Length of tonsillar descent (mm)** | 12.5 ± 6.12 | 7.8 ± 7.93 | 0.130[b] |
| **C-C2SVA ≥5** | 11 (45.8%) | 4 (36.4%) | 0.594[a] |
| **Length of C-C2SVA (mm)** | 4.8 (2.59) | 3.3 (3.40) | 0.291[b] |
| | 4.0 [3.0, 6.0] | 3.0 [0.0, 6.0] | |
| **pBC-2 ≥9 mm** | 9 (37.5%) | 2 (18.2%) | 0.349[a] |
| **Length of pBC-2 (mm)** | 7.4 (2.09) | 7.1 (1.31) | 0.418[b] |
| | 7.0 [6.0, 9.0] | 7.0 [6.0, 8.0] | |
| **CXA <125 degrees** | 3 (12.5%) | 0 (0.0%) | 0.382[a] |
| **Size of CXA (degrees)** | 138.1 (13.22) | 149.3 (10.77) | 0.016[b] |
| | 138.0 [130.0, 144.0] | 148.0 [141.0, 157.0] | |
| **Thickness of PAAM (mm)** | 2.6 (0.84) | 3.4 (1.99) | 0.376[b] |
| | 2.6 [1.9, 3.2] | 2.8 [2.0, 4.7] | |
| **Thickness of PAOM (mm)** | 2.7 (0.90) | 2.8 (0.72) | 0.740[b] |
| | 2.4 [2.0, 3.4] | 2.9 [2.4, 3.3] | |
| **Volume of posterior fossa (mm³)[d]** | 163,234 ± 20,142 | 218,305 ± 26,953 | **<0.001** |
| **Ratio posterior fossa volume/height[§]** | 129.9 ± 24.1 | 160.9 ± 28.22 | **0.028** |
| **Pathological Characteristics** | | | |
| **Collagen (%)** | 77.1 (18.65) | 85.5 (9.86) | 0.251[b] |
| | 80.0 [65.0, 90.0] | 90.0 [80.0, 95.0] | |
| **Fat (%)** | 22.9 (18.65) | 14.5 (9.86) | 0.251[b] |
| | 20.0 [10.0, 35.0] | 10.0 [5.0, 20.0] | |
| **Disorganization score** | | | **0.012[c]** |
| None | 1 (4.2%) | 0 (0.0%) | |
| Mild | 5 (20.8%) | 7 (63.6%) | |
| Moderate | 7 (29.2%) | 4 (36.4%) | |
| Severe | 11 (45.8%) | 0 (0.0%) | |
| **PAOM thickness (μm)** | 2031.3 (613.76) | 2363.6 (205.05) | 0.142[b] |
| | 2125.0 [1375.0, 2500.0] | 2500.0 [2250.0, 2500.0] | |

C-C2SVA, condylar to C2 sagittal vertical axis; pBC-2, perpendicular basion-inferior C2 distance; CXA, clival-axial angle; PAAM, posterior atlantoaxial membrane; PAOM, posterior atlanto-occipital membrane

[a] Fisher's exact test with Lancaster's mid-p correction.

[b] Wilcoxon rank-sum test.

[c] Cochran-Armitage trend test.

[d] Volume calculations were performed on 13 Chiari patients and 10 control patients.

Additional comparisons of PAOM disorganization, membrane thickness, and imaging measurements of PAOM thickness, as well as posterior fossa volume and volume/height ratio between groups did not reveal any significant correlations.

## Discussion

In this study, we analyzed the PAOM in children with and without CM-I and found that the membrane is more disorganized in patients with CM-I. This is the first anatomic study of this type in children and provides a frame of reference using control specimens.

**Table 5. Imaging and pathological/morphological characteristics comparison by sex in patients with Chiari I malformation.**

| Variable | Female (n = 12) | Male (n = 12) | Overall (n = 24) | p-value |
|---|---|---|---|---|
| *Imaging Characteristics* | | | | |
| **Presence of syrinx** | 2 (16.7%) | 3 (25.0%) | 5 (20.8%) | 0.820[a] |
| **Tonsillar descent (mm)** | 12.6 (6.16) | 12.4 (6.35) | 12.5 (6.12) | 0.862[b] |
| | 12.0 [8.9, 18.0] | 11.5 [7.0, 16.0] | 11.5 [8.4, 16.5] | |
| **C-C2SVA ≥5 mm** | 5 (41.7%) | 6 (50.0%) | 11 (45.8%) | 0.707[a] |
| **C-C2SVA (mm)** | 5.0 (2.94) | 4.6 (2.31) | 4.8 (2.59) | 0.907[b] |
| | 4.0 [3.0, 6.0] | 4.5 [3.0, 5.5] | 4.0 [3.0, 6.0] | |
| **pBC-2 ≥9 mm** | 6 (50.0%) | 3 (25.0%) | 9 (37.5%) | 0.245[a] |
| **pBC-2 (mm)** | 7.8 (2.20) | 7.1 (2.02) | 7.4 (2.09) | 0.465[b] |
| | 8.0 [6.0, 10.0] | 7.0 [6.5, 8.5] | 7.0 [6.0, 9.0] | |
| **CXA <125 degrees** | 2 (16.7%) | 1 (8.3%) | 3 (12.5%) | 0.609[a] |
| **CXA (degrees)** | 139.8 (15.20) | 136.4 (11.33) | 138.1 (13.22) | 0.954[b] |
| | 137.5 [132.5, 144.5] | 139.0 [127.5, 144.0] | 138.0 [130.0, 144.0] | |
| **PAAM (mm)** | 2.6 (0.59) | 2.7 (1.05) | 2.6 (0.84) | 0.877[b] |
| | 2.4 [2.2, 3.2] | 2.8 [1.9, 3.3] | 2.6 [1.9, 3.2] | |
| **PAOM (mm)** | 2.6 (0.93) | 2.7 (0.91) | 2.7 (0.90) | 0.600[b] |
| | 2.1 [1.9, 3.6] | 2.7 [2.2, 3.4] | 2.4 [2.0, 3.4] | |
| *Pathological Characteristics* | | | | |
| **Collagen (%)** | 76.3 (20.13) | 77.9 (17.90) | 77.1 (18.65) | 0.836[b] |
| | 85.0 [65.0, 90.0] | 80.0 [65.0, 92.5] | 80.0 [65.0, 90.0] | |
| **Fat (%)** | 23.8 (20.13) | 22.1 (17.90) | 22.9 (18.65) | 0.836[b] |
| | 15.0 [10.0, 35.0] | 20.0 [7.5, 35.0] | 20.0 [10.0, 35.0] | |
| **Disorganization score** | | | | 0.363[c] |
| None | 1 (8.3%) | 0 (0.0%) | 1 (4.2%) | |
| Mild | 3 (25.0%) | 2 (16.7%) | 5 (20.8%) | |
| Moderate | 3 (25.0%) | 4 (33.3%) | 7 (29.2%) | |
| Severe | 5 (41.7%) | 6 (50.0%) | 11 (45.8%) | |
| **PAOM thickness (μm)** | 2166.7 (567.42) | 1895.8 (652.43) | 2031.3 (613.76) | 0.214[b] |
| | 2500.0 [1750.0, 2500.0] | 2000.0 [1250.0, 2375.0] | 2125.0 [1375.0, 2500.0] | |

C-C2SVA, condylar to C2 sagittal vertical axis; pBC-2, perpendicular basion-inferior C2 distance; CXA, clival-axial angle; PAAM, posterior atlantoaxial membrane; PAOM, posterior atlanto-occipital membrane

[a]Fisher's exact test with Lancaster's mid-p correction.

[b]Wilcoxon rank-sum test.

[c]Cochran-Armitage trend test.

The PAOM is encountered during any posterior approach to the CCJ, but interestingly, there is no mention of this structure in *Gray's Anatomy* [6]. Previously, it was unknown whether there were differences in size (height, width, thickness) or histological architecture in patients with differing posterior fossa pathologies. We found that the membrane was more disorganized in children with CM-I, which may represent an accommodation rather than a pathological contribution. We also examined whether there were histopathological differences between the sexes and found no significant differences. The role of the PAOM in the pathogenesis and severity of CM-I and association with syringomyelia size or extent was not understood previously.

Our findings complement those of Alabaster et al. [4], whose investigation of the PAOM in adult patients with and without CM-I demonstrated that the ligamentous tissue is generally

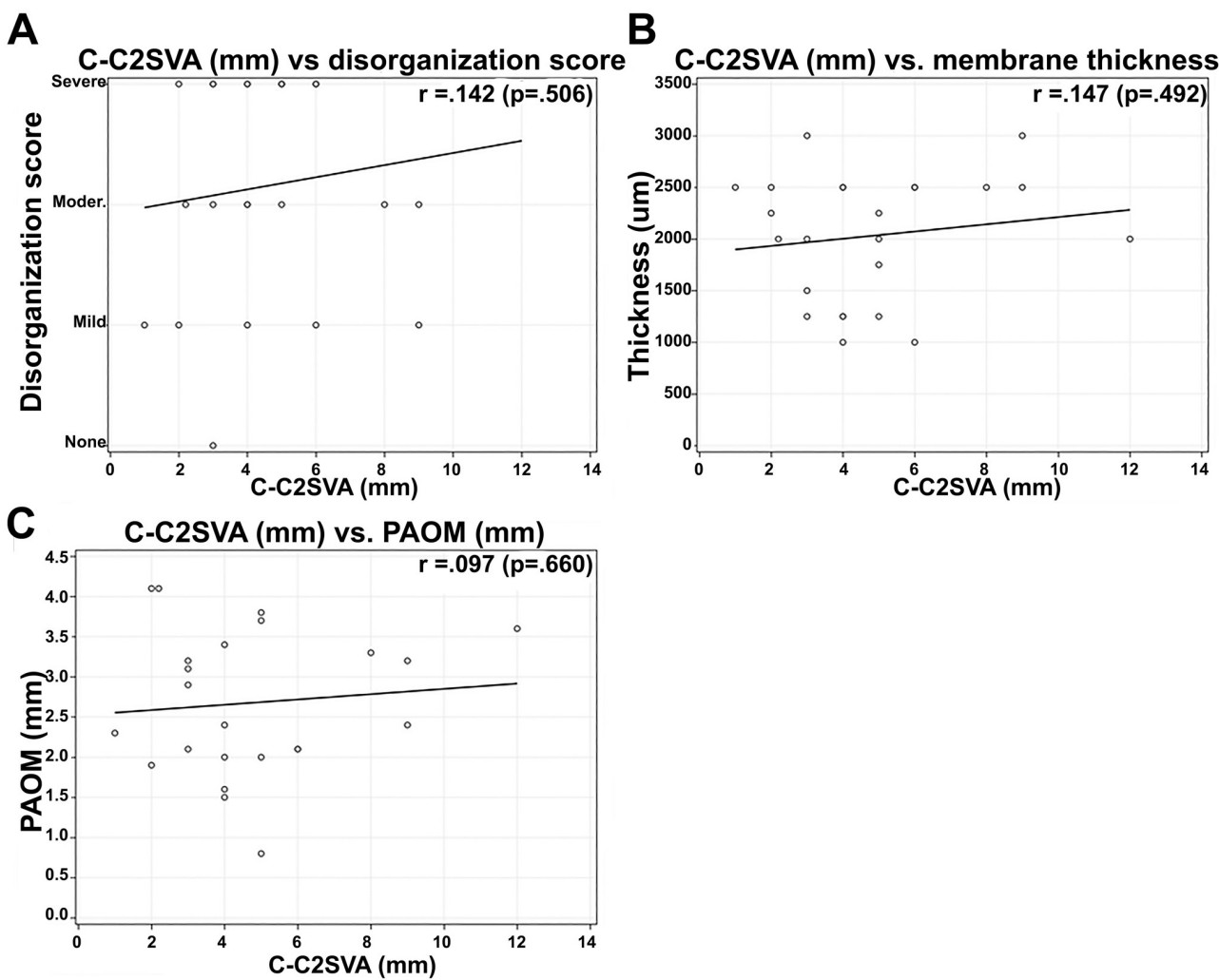

**Fig 6. Correlation analysis between C-C2SVA and pathological results of disorganization score (A), membrane thickness (B), and imaging measures of PAOM thickness (C).**

more disorganized—with poorly arranged collagen bands and interspersed adipose tissue—in patients with CM-I. We have now demonstrated that children with CM-I present with disorganized tissue similar to their adult counterparts. In adult CM-I specimens, Alabaster et al. [4] demonstrated more hyalinized fibrosis and directionally varied fibers with grade 1+ calcifications, whereas the specimens from patients without CM-I demonstrated uniformly horizontal fibers. Alabaster and colleagues did not observe hypertrophy of the suboccipital ligament in patients with CM-I, which contradicted previous reports. They hypothesized that the "appearance" of a hypertrophied ligament is based on the finding that patients with CM-I have a smaller posterior fossa [7–10]. Our set of specimens did not reveal any fiber splitting, fibrosis, or calcifications, indicating that these may occur more frequently in patients diagnosed with CM-I as adults than in patients diagnosed as children.

Using a 3D segmentation technique, we demonstrated that children with CM-I who undergo surgery have significantly smaller posterior fossa volume and volume/height ratio than those with posterior fossa tumor undergoing surgery. Previous reports have

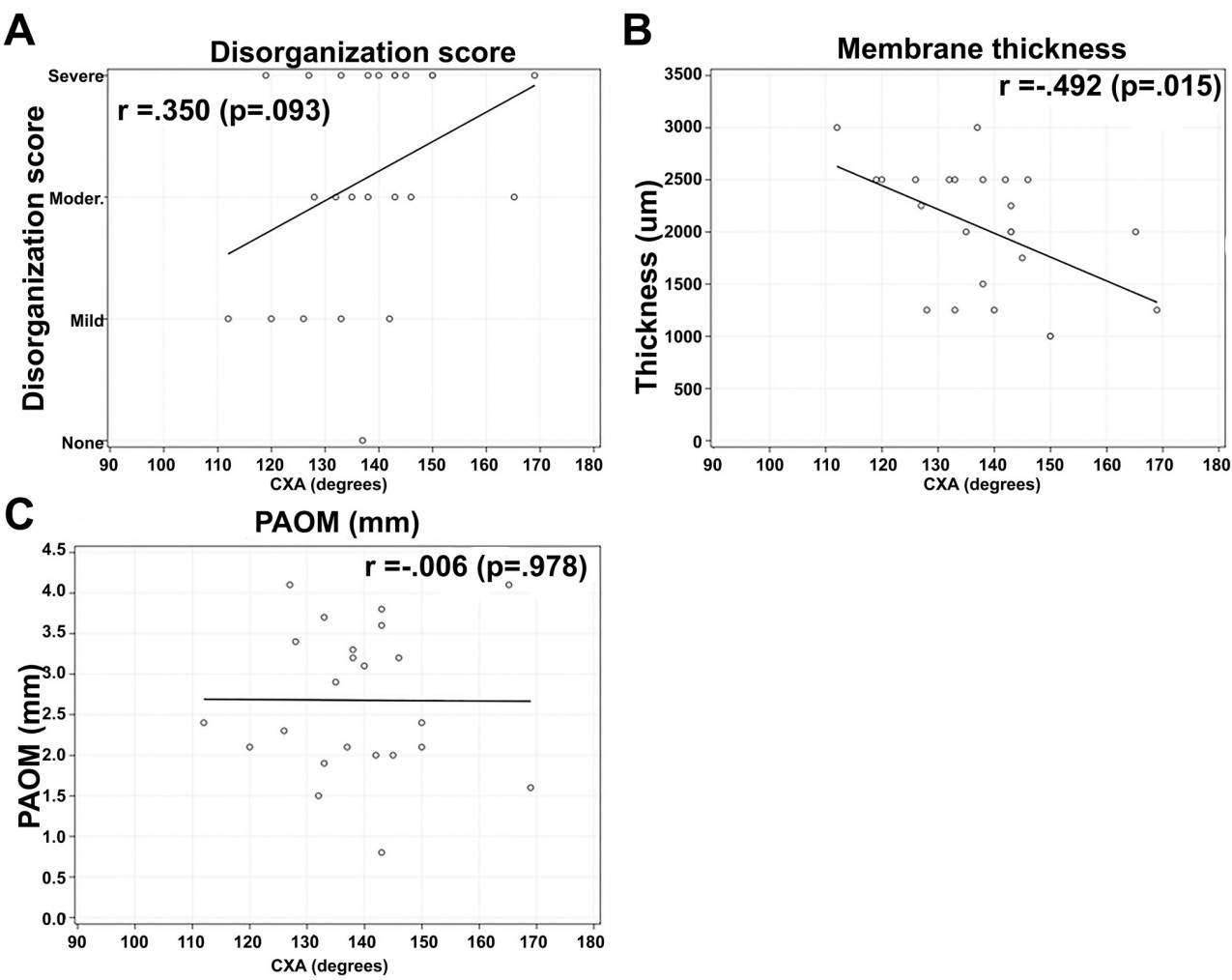

**Fig 7. Correlation analysis between CXA and pathological results of disorganization score (A), membrane thickness (B), and imaging measures of PAOM thickness (C).**

demonstrated this characteristic [11], which suggests that it is a causative mechanism of symptoms in CM-I, and volume expansion is a marker for clinical improvement after decompression [12]. However, a more disorganized PAOM structure in the setting of smaller posterior fossa volume suggests an acquired disorganization or thinning of the membrane as a mechanism to improve flow of CSF near the CCJ. In patients with CM-I, the posterior fossa compartment is small [13], which is what clinicians posit leads to the clinical presentation—headaches, syringomyelia, bulbar symptoms, etc.—of children with the malformation. The small posterior fossa leads to crowding at the foramen magnum with associated hyperdynamic local CSF pulsation with nonlaminar flow at the foramen magnum [14–16]. The surgical treatment for symptomatic CM-I involves expansion of the space at the foramen magnum via opening of the bone (suboccipital bone, C1 lamina), dura, and posterior atlanto-occipital membrane. Based on the findings of this study, the thinning and disorganization of the PAOM suggests an acquired mechanism or adaptation that is a physiologic response attempting to compensate for the malformation with auto-expansion of the posterior compartment. As a result, we

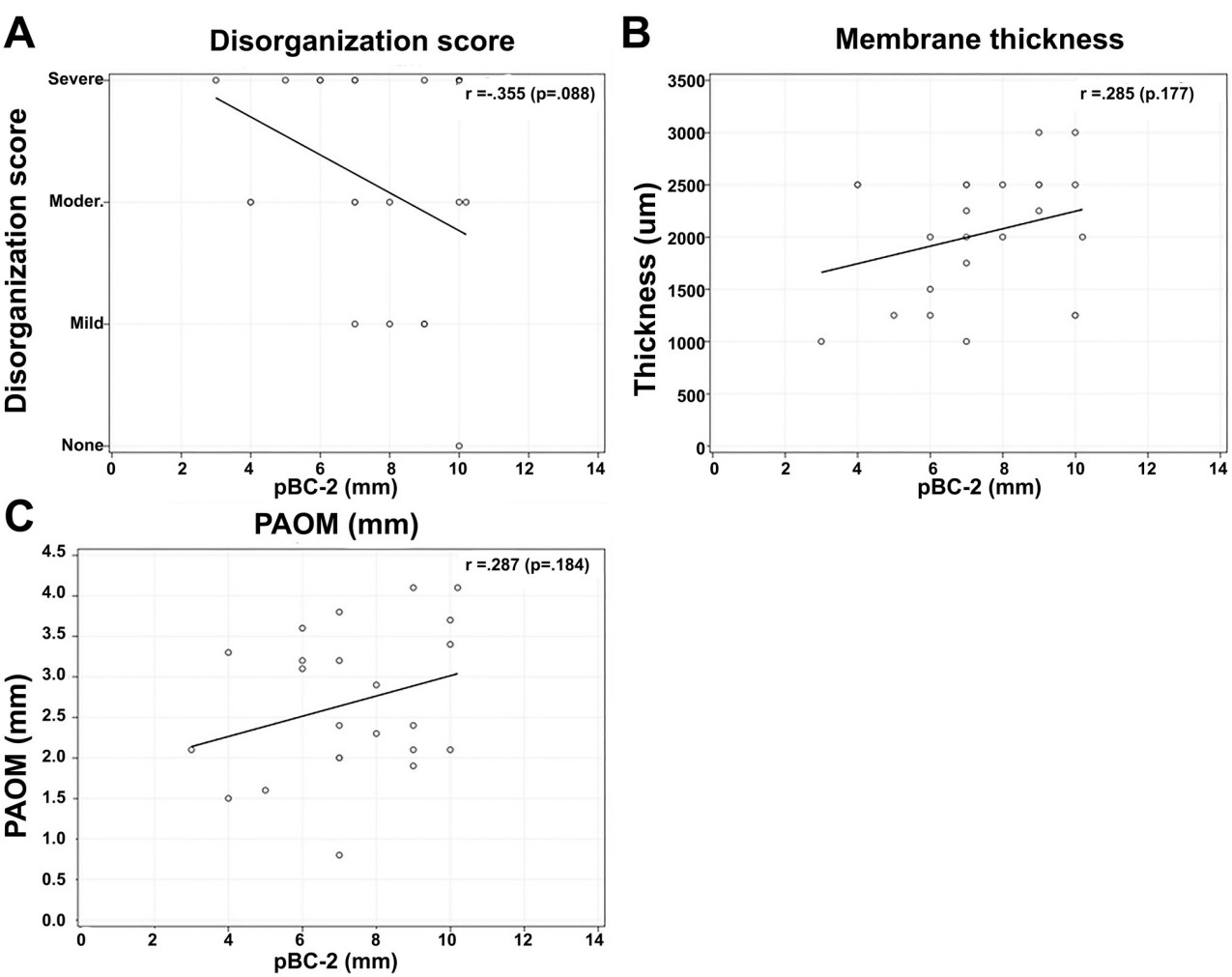

**Fig 8. Correlation analysis between pBC2 and pathological results of disorganization score (A), membrane thickness (B), and imaging measures of PAOM thickness (C).**

postulate that the membrane thins and becomes more disorganized with increased fat content (a chronic sign of tissue fatigue).

This finding suggests an adaptation rather than a causative factor. Interestingly, there was a lack of statistical correlation between membrane characteristics and posterior fossa volume.

Previous reports have hypothesized that the PAOM is hypertrophic in patients with CM-I and contributes to the ligament's subsequent pathology [7, 8, 17–32]. On the basis of its anatomic attachments to the occipital bone, C1 lamina, and bilateral condylar joints, the PAOM ligament may serve as a physiological "suspension system" for the cranial dural compartment by contributing to the rotational stability of the CCJ [4]. There are several reports of "immediate" enlargement of the CCJ stenosis in patients with CM-I after incision of the ligament [24, 33–38].

Nakamura et al. [39] examined material taken from the 'outer layer' of thickened dura mater (i.e., dural band) at the craniovertebral junction of 8 patients with syringomyelia and CM-I and found that the dural band was thickened and there were more collagen fibers that showed fiber splitting, hyaline nodules, calcification, and/or ossification, which were not

observed in 4 control patients. They suggested that the thickening of the dura mater might be a causative factor of syringomyelia with CM-I and that the histology of the thickened dura mater suggests the condition may be a consequence of birth injury in these patients. Although we did not study the dura mater here, the PAOM membrane findings suggest a relative disorganization and thinning of the membrane as a compensation for the turbulent hydrodynamics and biomechanics at the CCJ in patients with CM-I. We also found no evidence of hyaline nodules or calcification in pediatric specimens.

## Limitations

There are limitations to the current investigation. Although no specimens were excluded because of cautery artifact, there is inherent variability in the size of the specimen removed depending on the patient's surgical anatomy. Although we used a group of children with posterior fossa tumors as control patients, this is not an entirely typical cohort of children. We recognize there are limitations in comparing these children, who can have tonsillar ectopia, syringomyelia, and obstructive hydrocephalus. The rationale for using this cohort for comparison is that their tissue was readily available and removed as part of the standard surgical treatment. An ideal substrate for control data would be children without disease at the CCJ, but ethical standards prevent their use.

The measurement of anatomical compartmental volume in a patient with a posterior fossa neoplasm may represent an overestimation. Chiari malformation is a congenital anomaly except in cases where a secondary malformation develops from a tumor, CSF leak, etc. On the other hand, posterior fossa tumor/mass lesion growth that occurs after development of the compartments of the cranium is not congenital. Thus, we would expect the volumetric measurements of the posterior fossa to be a reasonable proxy for "normal," despite not being ideal. For this study, the goal of calculating posterior fossa volume was to compare the histological measures to the volumetric measure to correlate the specimens to the imaging findings. A third population of non-pathological controls would not allow us to achieve this goal. Additionally, non-pathological controls with imaging in the setting of traumatic brain injury/concussion/epilepsy do not routinely have the three-dimensional, fine-cut, high-resolution sequences available that are necessary to calculate posterior fossa volume. Had we included non-pathological control patients, additional recruitment and imaging studies outside the scope of the current study would have been necessary.

The outcome/variable of membrane disorganization was used to describe the structure of the PAOM in a clinically relevant manner. The ordinal disorganization index was derived by a senior, fellowship-trained neuropathologist and used with clear definitions and criteria. Each of the specimens was independently reviewed by two fellowship-trained neuropathologists who tested the method and consistently arrived at consensus. Validation of this method is needed moving forward.

This investigation was limited by the low number of participants—specifically control patients. Additionally, although there were differences seen between the groups, many were not statistically significant, which could be the result of a low number of subjects/specimens. The patients were not matched so as to maximize the number of subjects/participants in the study. Although this was an exploratory study, there is no minimal clinically important difference known for the pathological variables investigated. Larger-scale study of the phenomena discovered is warranted. The population was racially homogeneous, which may limit the generalizability of these pathological findings across broader populations. Future study will investigate molecular features of the PAOM, bone, dura, and surrounding muscle tissue to further understand this anatomically unique and complex region. Although this study is not likely to

impact clinical practice directly, the PAOM may represent a target for future therapies as a mechanism of improving CSF flow pulsation at the craniocervical junction. Despite these limitations, the information in this report is of interest in understanding the pathophysiologic mechanism of CM-I and potentially treating patients with CM-I.

## Conclusions

In patients with CM-I, the PAOM demonstrates a more disorganized architecture than in patients without CM-I but no significant difference in thickness. This likely represents an anatomic adaptation in the setting of CM-I rather than a pathologic contribution. Further study of the interplay between the PAOM, dura, and bone structures in this region will be necessary to fully understand their relationship in patients with CM-I.

## Supporting information

**S1 Data. Containing all demographic and medical information used to study Chiari I malformation and control cohorts.**
(XLSX)

## Acknowledgments

We thank Kristin Kraus and Cortlynd Olsen for editorial assistance in preparing this paper.

## Author Contributions

**Conceptualization:** Vijay M. Ravindra, Douglas L. Brockmeyer.

**Formal analysis:** Hailey Jensen, Russell Telford, Justin Ryan.

**Investigation:** Lorraina Robinson, Elena Kurudza, Evan Joyce, Osama Youssef, Justin Ryan, Robert J. Bollo, Rajiv R. Iyer, John R. W. Kestle, Samuel H. Cheshier, Qinwen Mao, Douglas L. Brockmeyer.

**Methodology:** Vijay M. Ravindra, Douglas L. Brockmeyer.

**Project administration:** Allison Ludwick, Daniel S. Ikeda.

**Writing – original draft:** Vijay M. Ravindra, Lorraina Robinson.

**Writing – review & editing:** Vijay M. Ravindra, Hailey Jensen, Elena Kurudza, Russell Telford, Justin Ryan, Robert J. Bollo, Rajiv R. Iyer, John R. W. Kestle, Samuel H. Cheshier, Daniel S. Ikeda, Douglas L. Brockmeyer.

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
