## [Decision Letter · Decision Letter 0]

23 Oct 2023

PONE-D-23-31525

Morphological and ultrastructural investigation of the posterior atlanto-occipital membrane: comparing children with Chiari malformation type I and controls

PLOS ONE

Dear Dr. Ravindra,

Thank you for submitting your manuscript to PLOS ONE. After careful consideration, we feel that it has merit but does not fully meet PLOS ONE’s publication criteria as it currently stands. Therefore, we invite you to submit a revised version of the manuscript that addresses the points raised during the review process.

We look forward to receiving your revised manuscript.

Kind regards,

Sagar Panthi, MBBS

Academic Editor

PLOS ONE

Journal Requirements:

   "The American Syringomyelia & Chiari Alliance Project Inc. (ASAP) funded the project for a sum of $19,982.00 as part of the Timothy M. George Fellowship Award to Vijay Ravindra."

5. Please amend the manuscript submission data (via Edit Submission) to include author  John R.W. Kestle, MD. 

Additional Editor Comments:

Please check the following details to meet the journal's publication criteria. If anything is unaddressed, please make the necessary address in the subsequent revisions: 

1. Financial Disclosure:

Enter a statement with the following details: Initials of the authors who received each award • • Grant numbers awarded to each author • The full name of each funder • **URL of each funder website**

**Did the sponsors or funders play any role in the study design, data collection and analysis, decision to publish, or preparation of the manuscript? **

2. Ethical Clearance: 

Please clarify in the ethical approval section that the hospital is affiliated to the university which approved the study as the hospital's name is not clearly indicated in the approval letter.

3. Figure captions

Figure captions must be inserted in the text of the manuscript, immediately following the paragraph in which the figure is first cited (read order). Do not include captions as part of the figure files themselves or submit them in a separate document.

Reviewers' comments:

Reviewer's Responses to Questions

**Comments to the Author**

1. Is the manuscript technically sound, and do the data support the conclusions?

Reviewer #1: Partly

Reviewer #2: Yes

2. Has the statistical analysis been performed appropriately and rigorously? 

Reviewer #1: Yes

Reviewer #2: Yes

3. Have the authors made all data underlying the findings in their manuscript fully available?

Reviewer #1: Yes

Reviewer #2: No

4. Is the manuscript presented in an intelligible fashion and written in standard English?

Reviewer #1: Yes

Reviewer #2: Yes

5. Review Comments to the Author

Reviewer #1: The authors performed a study of children undergoing posterior fossa craniotomy for Chiari 1 malformation or brain tumor resection. They then compared the PAOM between these two groups. Differences between Chiari and tumor were seen in posterior fossa volume and disorganization score, with Chiari patients showing higher degree of disorganization. There were no significant associations between most imaging measurements and the collagen disorganization score.

Recruitment was 9/21 to 6/23. Why were these dates chosen? How was sample size determined for this prospective study? There were differences seen between the groups, but many were not statistically significant. Is this because the sample is too small? What would be considered a clinically relevant difference in the key variables?

The primary outcome seems to be the score indicating the degree of disorganization of the collagen (0 to 3). Has this score been validated? Any testing of reliability?

How likely is it that the difference in posterior fossa volume observed here is related to the presence of a mass in the posterior fossa? This seems an odd control group for the volume measurement. Non-pathological controls might be more appropriate.

In the Discussion and Conclusion, the authors state that their findings suggest that the collagen findings are more likely to be an adaptation rather than a cause of Chiari pathology. It is not clear what they base this statement on? How to the present results support this conclusion? This point is made in several places, including the Abstract. More justification of this statement is needed, or the conclusion should be revised.

Reviewer #2: 1. A well written article. However the sample size is small and the cases and controls are unevenly distributed. A total of 35 children with 11 as controls. Were they matched? Pls explain.

2. As the number of patients were few, I suggest the authors to list all 24 patients with age sex, preop imaging findings, post op complications and histological findings.

3. Pls describe in the discussion section, the implication of this research to clinical practice. Will it affect the way we perform surgery or any medicines you theorize to address the disorganized membrane?

6. PLOS authors have the option to publish the peer review history of their article (what does this mean?). If published, this will include your full peer review and any attached files.

Reviewer #1: No

Reviewer #2: No

---

## [Author Response · Author response to Decision Letter 0]

9 Nov 2023

November 1, 2023

Sagar Panthi, MBBS

Academic Editor

PLOS ONE

Dear Dr. Panthi:

Thank you for sending us the reviewers’ comments regarding our paper, "Morphological and ultrastructural investigation of the posterior atlanto-occipital membrane: comparing children with Chiari malformation type I and controls” (PONE-D-23-31525). We are very pleased that the reviewers found the paper to be of interest. In response to their comments, we have made the following changes to the paper. 

RESPONSE TO REVIEWERS

Reviewer #1

1. The authors performed a study of children undergoing posterior fossa craniotomy for Chiari 1 malformation or brain tumor resection. They then compared the PAOM between these two groups. Differences between Chiari and tumor were seen in posterior fossa volume and disorganization score, with Chiari patients showing higher degree of disorganization. There were no significant associations between most imaging measurements and the collagen disorganization score.

Recruitment was 9/21 to 6/23. Why were these dates chosen? How was sample size determined for this prospective study? There were differences seen between the groups, but many were not statistically significant. Is this because the sample is too small? What would be considered a clinically relevant difference in the key variables?

RESPONSE: Recruitment for the study was based on the study timeline proposed as part of the funding mechanism. The original study was set to be completed over a 12-month period with the intention of examining and analyzing 40 specimens. Because of the COVID-19 pandemic, we had a difficult time enrolling patients and, therefore, obtaining specimens. The study was extended to June 2023, but the final enrollment was only 35 patients. 

This is the first study of its type, and therefore there was no specific primary outcome measure. Instead, we performed a comparative analysis of children with and without CM-I that required surgical treatment. Based on the findings of this exploratory study, there is no obvious clinically relevant difference per se. Additionally, no metric exists for disorganization scoring or thickness with respect to minimal clinically important difference. The results of this study, do, however, shed light on the potential pathophysiologic mechanism of CM-I and its effect on other mesodermal structures, in this case the PAOM. The sample size is small and a larger-scale study of the phenomena is warranted.

As a result of the reviewer’s request, we have added this to the limitations section of the paper (page 24).

2. The primary outcome seems to be the score indicating the degree of disorganization of the collagen (0 to 3). Has this score been validated? Any testing of reliability?

RESPONSE: The outcome/variable of membrane disorganization was used to describe the structure of the PAOM in a clinically relevant manner. Initially, percentage of fat content and fiber organization were considered for use, but we determined that describing the structure that way would have very little meaning to the readership or clinicians. Because, to the knowledge of the authors, no published criteria for organization/disorganization is available, the ordinal disorganization index was derived by a senior, fellowship-trained neuropathologist and used with clear definitions and criteria. Each specimen was independently reviewed by two fellowship-trained neuropathologists who tested the method and consistently arrived at consensus; however, the grading system has not been formally validated. This information has been added to the methods (page 8) and limitations (page 23) sections of the paper.

3. How likely is it that the difference in posterior fossa volume observed here is related to the presence of a mass in the posterior fossa? This seems an odd control group for the volume measurement. Non-pathological controls might be more appropriate.

RESPONSE: The use of posterior fossa tumor patients as controls is a limitation, but for a surgical pathology study there is no other comparison with respect to membrane histology/ultrastructural properties. 

Chiari malformation is a congenital anomaly except in cases where a secondary malformation develops from a tumor, CSF leak, etc. On the other hand, posterior fossa tumor/mass lesion growth that occurs after development of the compartments of the cranium is not congenital. Thus, we would expect the volumetric measurements of the posterior fossa in patients with tumor/mass lesion growth to be a reasonable proxy for “normal,” despite not being ideal. In calculating posterior fossa volume, the goal was to compare the histological measures to the volumetric measure to correlate the specimens to the imaging findings. A third population of non-pathological controls would not allow us to achieve this goal. Additionally, non-pathological controls with imaging in the setting of TBI/concussion/epilepsy would not have the three-dimensional, fine-cut, high-resolution sequences available that are necessary to calculate posterior fossa volume. Had we included non-pathological control patients, additional recruitment and imaging studies would have been needed, which were outside the scope of and exceeded the financial constraints of the current study. 

The limitations of the use of posterior fossa tumor patients as the control population has been added to the paper (page 23). 

4. In the Discussion and Conclusion, the authors state that their findings suggest that the collagen findings are more likely to be an adaptation rather than a cause of Chiari pathology. It is not clear what they base this statement on? How to the present results support this conclusion? This point is made in several places, including the Abstract. More justification of this statement is needed, or the conclusion should be revised.

RESPONSE: In patients with CM-I, the posterior fossa compartment is small (a), which is part of what leads to the clinical presentation—headaches, syringomyelia, bulbar symptoms, etc.—of children with the malformation. The small posterior fossa leads to crowding at the foramen magnum with associated hyperdynamic local CSF pulsation with nonlaminar flow at the foramen magnum (b,c,d). The surgical treatment for symptomatic CM-I involves expansion of the space at the foramen magnum via opening of the bone (suboccipital bone, C1 lamina), dura, and posterior atlanto-occipital membrane. Based on the findings of this study, the thinning and disorganization of the PAOM is an acquired mechanism or adaptation that is a physiologic response attempting to compensate for the malformation with auto-expansion of the posterior compartment. As a result, we posit that the membrane thins and becomes more disorganized with increased fat content (a chronic sign of tissue fatigue). This information, including references is included in the discussion portion of the paper (page 21). 

a. Botelho RV, Heringer LC, Botlho PB, Lopes RA, Waisberg J. Posterior fossa dimensions in chiari malformation patients compared with normal subjects: systematic review and meta-analysis. World Neurosurg. 2020 Jun: 138: 521-529.

b. Heiss JD. Cerebrospinal fluid hydrodynamics in chiari i malformation and syringomyelia: modeling pathophysiology. Neurosurg Clin N Am. 2023 Jan; 34(1): 81-90. 

c. Chiari H. Ueber Veränderungen des Kleinhirns infolge von Hydrocephalie des Grosshirns. Dtsch Med Wochenschr 1891;17:1172–5. 

d. Wilkins RH, Brody IA. The Arnold-Chiari malformation. Neurological classics 38. Arch Neurol 1971;25:376–9. 

Reviewer #2:

1. A well written article. However the sample size is small and the cases and controls are unevenly distributed. A total of 35 children with 11 as controls. Were they matched? Pls explain.

RESPONSE: We thank the reviewer for their compliment. We recognized that the sample size is small with uneven distribution of cases and controls. As noted in response 1 to Reviewer #1, patient enrollment was interrupted by the COVID-19 pandemic and recruitment goals were necessarily altered. Given the desire to recruit as many subjects/pathological specimens as possible, the patients were not matched. Although disparities in body mass index, sex, and age were a concern, our post hoc analysis (Table 1) did not reveal any differences in age, sex distribution, or BMI. Thus, we believe a relative comparison is reasonable for this anatomic structure analysis. We have added the lack of formal matching to the limitations portion of the paper (page 24). 

2. As the number of patients were few, I suggest the authors to list all 24 patients with age sex, preop imaging findings, post op complications and histological findings.

RESPONSE: We appreciate the suggestion, however because of the homogeneity in age and sex as well as histopathological findings and the lack of postoperative complications within the CM-I cohort we do not think that a table with this information would add to the paper. 

3. Pls describe in the discussion section, the implication of this research to clinical practice. Will it affect the way we perform surgery or any medicines you theorize to address the disorganized membrane?

RESPONSE: Please see responses 1 and 4 to Reviewer #1 above. Our findings further the current understanding of the pathophysiology of CM-I and the role of the PAOM. This study is not likely to impact clinical practice directly, but thinning of the membrane as an adaptive response may represent a potential for therapeutic targets in the future to locally improve the flow of CSF and pulsatility at the foramen magnum (i.e., agents administered directly or indirectly to thin the membrane). We have added this to the discussion section of the paper, but we have kept this insertion brief so as not to overstate the clinical implications of our results (page 24). 

We thank the editor for the opportunity to revise the paper and believe that with the aforementioned changes it is significantly improved. 

We hope it meets the standard for publication in PLoS One and look forward to your decision.

---

## [Decision Letter · Decision Letter 1]

16 Nov 2023

PONE-D-23-31525R1Morphological and ultrastructural investigation of the posterior atlanto-occipital membrane: comparing children with Chiari malformation type I and controlsPLOS ONE

Dear Dr. Ravindra,

Thank you for submitting your manuscript to PLOS ONE. After careful consideration, we feel that it has merit but does not fully meet PLOS ONE’s publication criteria as it currently stands. Therefore, we invite you to submit a revised version of the manuscript that addresses the points raised during the review process.

We look forward to receiving your revised manuscript.

Kind regards,

Sagar Panthi, MBBS

Academic Editor

PLOS ONE

Journal Requirements:

Additional Editor Comments:

Thank you to the authors for sending us with the revisions addressing the suggestions and queries as asked by the reviewers. There are still a few suggestions from Reviewer 1 which needs addressing. In addition, please also address the suggestions made by the editorial team in the previous revision.

Reviewers' comments:

Reviewer's Responses to Questions

**Comments to the Author**

1. If the authors have adequately addressed your comments raised in a previous round of review and you feel that this manuscript is now acceptable for publication, you may indicate that here to bypass the “Comments to the Author” section, enter your conflict of interest statement in the “Confidential to Editor” section, and submit your "Accept" recommendation.

Reviewer #1: All comments have been addressed

Reviewer #2: All comments have been addressed

2. Is the manuscript technically sound, and do the data support the conclusions?

Reviewer #1: Yes

Reviewer #2: Yes

3. Has the statistical analysis been performed appropriately and rigorously? 

Reviewer #1: Yes

Reviewer #2: N/A

4. Have the authors made all data underlying the findings in their manuscript fully available?

Reviewer #1: Yes

Reviewer #2: Yes

5. Is the manuscript presented in an intelligible fashion and written in standard English?

Reviewer #1: Yes

Reviewer #2: Yes

6. Review Comments to the Author

Reviewer #1: The authors have addressed the reviewer comments.

One additional suggestion:

Recommend removing the final statement in conclusion regarding refutation of prior hypotheses. Because the abstract does not make reference to these hypotheses, it is difficult to understand what is meant by this comment.

Also, in the Conclusion section, while the refuted hypotheses have now at least been mentioned in the Discussion, this statement is probably a bit too strong given the provided data. Removing that clause about refutation would do nothing to weaken this report.

Reviewer #2: The authors have satisfactorily addressed my queries and I approve this manuscript for publication in the PLOS ONE

7. PLOS authors have the option to publish the peer review history of their article (what does this mean?). If published, this will include your full peer review and any attached files.

Reviewer #1: No

Reviewer #2: **Yes: **Mohan Raj Sharma

---

## [Author Response · Author response to Decision Letter 1]

17 Nov 2023

We thank the reviewer and have made the suggested change and have removed the sentence about refutation of prior hypotheses from the conclusion section of the abstract and the conclusion of the manuscript itself. 

We agree the suggested removal does nothing to weaken the report.

We thank the editor for the opportunity to revise the paper and believe that with the changes it is improved. 

We hope it meets the standard for publication in PLoS One and look forward to your decision.

Sincerely,

Vijay M. Ravindra, MD, MSPH

---

## [Editor Report · Decision Letter 2]

10 Dec 2023

Morphological and ultrastructural investigation of the posterior atlanto-occipital membrane: comparing children with Chiari malformation type I and controls

PONE-D-23-31525R2

Dear Dr. Ravindra,

We’re pleased to inform you that your manuscript has been judged scientifically suitable for publication and will be formally accepted for publication once it meets all outstanding technical requirements.

Kind regards,

Sagar Panthi, MBBS

Academic Editor

PLOS ONE
---

## [Editor Report · Acceptance letter]

5 Jan 2024

PONE-D-23-31525R2 

PLOS ONE

Dear Dr. Ravindra, 

I'm pleased to inform you that your manuscript has been deemed suitable for publication in PLOS ONE. Congratulations! Your manuscript is now being handed over to our production team.

Kind regards, 

on behalf of

Dr. Sagar Panthi 

Academic Editor

PLOS ONE